# Unconventional conservation reveals structure-function relationships in the synaptonemal complex

Lisa E Kursel*, Henry D Cope, Ofer Rog*

School of Biological Sciences and Center for Cell and Genome Sciences, University of Utah, Salt Lake City, United States

**Abstract** Functional requirements constrain protein evolution, commonly manifesting in a conserved amino acid sequence. Here, we extend this idea to secondary structural features by tracking their conservation in essential meiotic proteins with highly diverged sequences. The synaptonemal complex (SC) is a ~100-nm-wide ladder-like meiotic structure present in all eukaryotic clades, where it aligns parental chromosomes and regulates exchanges between them. Despite the conserved ultrastructure and functions of the SC, SC proteins are highly divergent within *Caenorhabditis*. However, SC proteins have highly conserved length and coiled-coil domain structure. We found the same unconventional conservation signature in *Drosophila* and mammals, and used it to identify a novel SC protein in *Pristionchus pacificus*, Ppa-SYP-1. Our work suggests that coiled-coils play wide-ranging roles in the structure and function of the SC, and more broadly, that expanding sequence analysis beyond measures of per-site similarity can enhance our understanding of protein evolution and function.

## Editor's evaluation

Although the synaptonemal complex is an essential, deeply conserved structure that holds meiotic chromosomes together, the constituent proteins evolve exceptionally rapidly. This rapid evolution has hindered the identification of synaptonemal complex proteins based solely on sequence homology. This article overcomes this challenge by developing and validating a clever protein structure-based approach that leverages sequence divergence – rather than sequence conservation – to identify novel synaptonemal complex components.

*For correspondence:
lisa.kursel@utah.edu (LEK);
ofer.rog@utah.edu (OR)

**Competing interest:** The authors declare that no competing interests exist.

## Introduction

Functional and structural constraints leave evolutionary signatures on proteins. Often, functionally important domains undergo purifying selection and tend to evolve slowly. For example, enzymatic active sites require precise positioning of amino acids and can be identified based on sequence conservation. Even seeming exceptions are telling. Many genes in the immune system are fast-evolving and undergo recurrent changes (positive selection) to uphold tight interaction interfaces with foreign proteins (*Sawyer et al., 2005*; *Mitchell et al., 2012*; *Daugherty and Malik, 2012*).

This paradigm of protein evolution holds true for most studied proteins, but several exceptions have been identified (*Lewis and Wuttke, 2012*; *Wang et al., 2007*). One such example is the protein components of the synaptonemal complex (SC), and specifically, the central region of the SC (referred to throughout as 'the SC'; *Figure 1A*). The SC is present in all eukaryotic clades and is essential for meiosis. It brings parental chromosomes into close proximity in meiotic prophase and forms the interface between them. The SC also regulates genetic exchanges (crossovers), which serve as the

The online version of this article includes the following figure supplement(s) for figure 1:

**Source data 1.** Syntenic location of *Caenorhabditis* synaptonemal complex genes.

**Source data 2.** Multiple sequence alignments used in phylogenetic analysis.

**Source data 3.** Sequences of manually curated genes.

**Source data 4.** Multiple sequence alignments used in evolutionary analyses.

**Source data 5.** Sequences of *Caenorhabditis* synaptonemal complex proteins.

**Figure supplement 1.** Identification of *Caenorhabditis* synaptonemal complex proteins.

**Figure supplement 2.** Maximum likelihood phylogenies of *Caenorhabditis* synaptonemal complex proteins.

**Figure supplement 3.** Map of sites of under purifying, neutral, or positive selection.

**Figure 1.** The divergence of synaptonemal complex (SC) proteins is driven by neutral evolution. (**A**) Left: diagram of the SC. The N- and C-termini of transverse filaments are labeled. SC, pink; axis (scaffold for SC assembly), dark gray; chromosomes, light gray. Right: electron micrographs of the SC from nematode (*C. elegans*), mouse (*M. musculus*), and beetles (*B. cribrosa*), demonstrating the conserved organization and dimensions of the SC. Left image is reproduced from **Figure 1** of *Rog et al., 2017*, center image is reproduced from *Kouznetsova et al., 2011*, and right image is reproduced from **Figure 1** of *Schmekel et al., 1993*. Scale bars = 100 nm. (**B**) Abridged *Caenorhabditis* species tree. Presence of SC proteins is to the right of each species. Filled box, present; unfilled box, no ortholog identified. For full tree, see *Figure 1—figure supplement 1*. (**C**) Graph of average pairwise percent identity for SC proteins and SMC-1/3 (two chromatin-associated coiled-coil proteins) as controls. Colored nodes on the x-axis correspond to the species tree in (**B**). Evolutionary time increases from left to right with time estimates according to *Cutter, 2008* listed below select nodes. (**D**) Dot plot comparing amino acid substitutions per site of SC proteins to all other *Caenorhabditis* proteins. Black circle, median value. The median SC amino acid substitutions per site = 1.64, other = 0.43, Wilcoxon rank sum test p-value=0.0005. SMC-1/3 (green) and LEV-11 (purple) are highlighted as controls. The high divergence of SC proteins cannot be explained by positive selection (Table 1).

*Figure 1 continued on next page*

physical link between parental chromosomes during the first meiotic division. First observed more than 60 years ago (*Moses, 1956*; *Fawcett, 1956*), the SC is a 100-nm-wide, ladder-like structure with regularly spaced rungs (*Figure 1A*). In the decades since, electron microscopy allowed the characterization of the SC in meiocytes from numerous organisms, where its ultrastructure was found to be remarkably conserved (*Gillies and Moens, 2008*; *Carpenter, 1975*; *MacQueen et al., 2002*; *Schmekel et al., 1993*; *Moses et al., 1977*). The advent of molecular genetics allowed cloning of SC proteins and revealed that they are perplexingly divergent and cannot be identified across distant taxa based on sequence homology. In contrast, the components of the axis, another meiosis-specific chromosomal structure and the assembly platform for the SC, share clear sequence homology between clades as distant as plants and humans (*Rillo-Bohn et al., 2021*; *Cobbe and Heck, 2004*; *Aravind and Koonin, 1998*).

Despite poor per-site identity, in cases where SC proteins have been cloned at least one assumes a stereotypical head-to-head orientation spanning the space between the parental chromosomes: N-termini pointing toward the center of the SC and C-termini pointing outward (*Figure 1A*). SC proteins with this orientation are

referred to as 'transverse filaments' and help determine the width of the SC via a central coiled-coil domain (*Sym and Roeder, 1995*; *Tung and Roeder, 1998*; *Billmyre et al., 2019*; *Ollinger et al., 2005*). Thus, SC proteins represent a case where functional and ultrastructural conservation does not seem to constrain a protein's primary sequence – a pattern that, although unconventional for essential proteins, is likely to be more prevalent than is currently appreciated (*Woodruff, 2018*).

Here, we attempt to resolve this paradox of SC protein evolution. We find that SC proteins in *Caenorhabditis*, although highly divergent, harbor features that are strikingly conserved: protein length, as well as the length and location of coiled-coils. The conservation of these features is also exhibited by SC proteins in *Drosophila* and *Eutherian* mammals. We harness this intra-clade evolutionary signature to predict and identify a novel SC protein in the nematode *Pristionchus pacificus*. Our findings suggest that the fitness landscape of SC proteins is governed by secondary structures, shedding light on structure/function relationships of this conserved chromosomal interface.

## Results

### SC proteins are ancient and preserved in *Caenorhabditis*

To analyze the evolution of SC proteins, we generated and refined a dataset of all known SC proteins from 25 *Caenorhabditis* species. These species, many of which have been sequenced in the last two years, represent the *Elegans* and *Drosophilae* supergroups as well as two basally branching *Caenorhabditi*ds, *Caenorhabditis plicata* and *Caenorhabditis monodelphis* (*Figure 1B*, *Figure 1—figure supplement 1*). In the model organism *Caenorhabditis elegans,* six SC proteins have been identified: SYP-1–SYP-6 (*MacQueen et al., 2002*; *Hurlock et al., 2020*; *Colaiácovo et al., 2003*; *Smolikov et al., 2007*; *Smolikov et al., 2009*; *Zhang et al., 2020*). SYP-1, SYP-5, and the *C. elegans* -specific SYP-5 paralog, SYP-6, are transverse filament proteins (*MacQueen et al., 2002*; *Hurlock et al., 2020*; *Colaiácovo et al., 2003*; *Schild-Prüfert et al., 2011*; *Köhler et al., 2020*). All SC proteins are present in the *Elegans* and *Drosophilae* supergroups, indicating that they have been preserved for over 30 million years (*Cutter, 2008*; *Figure 1B*, *Figure 1—figure supplement 1*). This broad preservation is not surprising given that SC proteins in *C. elegans* are interdependent for their function and that their elimination causes a dramatic drop in viable progeny (*MacQueen et al., 2002*; *Hurlock et al., 2020*; *Colaiácovo et al., 2003*; *Smolikov et al., 2007*; *Smolikov et al., 2009*; *Zhang et al., 2020*). In addition, we found three instances of protein duplication; the previously identified paralogs SYP-5 and SYP-6 in *C. elegans* (*Hurlock et al., 2020*), a SYP-1 duplication in *Caenorhabditis panamensis* and a SYP-2 duplication in the common ancestor of *Caenorhabditis nouraguensis* and *Caenorhabditis becei* (*Figure 1B*, *Figure 1—figure supplements 1 and 2*).

To identify SC proteins in the early diverging *Caenorhabditis* species, *C. plicata* and *C. monodelphis,* we used all of our previously identified sequences as queries in BLASTP and tBLASTn searches, with a lenient e-value cutoff (1.0e$^{-1}$). This allowed us to identify SYP-3, -4, and -5 orthologs in both *C. plicata* and *C. monodelphis* (*Figure 1B*, *Figure 1—figure supplement 1*). We also found SYP-2 in *C. plicata* but not in *C. monodelphis*. We were unable to identify SYP-1 in either *C. plicata* or *C. monodelphis* (*Figure 1B*, *Figure 1—figure supplement 1*). It is possible that *C. plicata* and *C. monodelphis* have fewer SC proteins. However, given the fact that SC proteins are essential for meiosis and functionally interdependent, a plausible hypothesis is that the SC proteins are too diverged to be detected in these distantly related species. In line with this possibility, SC protein amino acid percent identity drops rapidly as more distantly related species are included in the comparison (*Figure 1C*).

### Neutral evolution drives the high divergence of *Caenorhabditis* SC proteins

The difficulty in identifying SC proteins in distantly related *Caenorhabditis* species is not surprising. *Drosophila* SC proteins were not found in other insects (*Hemmer and Blumenstiel, 2016*). Among vertebrates, sequence similarity of the transverse filament protein SYCP1 is limited to two short motifs, which are absent in *Caenorhabditis* and *Drosophila* (*Fraune et al., 2012*). We found that SC proteins are among the most diverged proteins in *Caenorhabditis*. On average, there are significantly more amino acid substitutions per site in SYP-1–SYP-5 compared to the *Caenorhabditis* proteome (SC proteins: median amino acid substitutions per site = 1.64, other proteins = 0.43, p-value=0.0005, *Figure 1D*). This divergence is not merely due to the prevalence of coiled-coils: the sequences of

**Table 1.** Summary table of tests for positive selection on synaptonemal complex proteins and SMC-1 and SMC-3 (controls) in the *Elegans* group species of *Caenorhabditis* (*Figure 1—figure supplement 1*).

p-values from likelihood ratio tests comparing CodeML models M1 vs. M2, M7 vs. M8, and M8a vs. M8 are listed. Each comparison tests the fit of the data to a model that does not allow positive selection (M1, M7, and M8a) to a model that does allow positive selection (M2 and M8). The numbers of sites under positive or negative selection in each protein according to the Fixed Effects Likelihood analysis from HyPhy with p-value<0.05 are also displayed.

| | | PAML CodeML | | | HyPhy fixed effects likelihood | |
|---|---|---|---|---|---|---|
| | Number of species | M1 vs. M2 p-value | M7 vs. M8 p-value | M8a vs. M8 p-value | Sites under positive selection | Sites under negative selection |
| SYP-1 | 12 | 1.00 | 0.99 | 0.60 | 0/451 | 177/451 (39%) |
| SYP-2 | 12 | 1.00 | 0.90 | 0.56 | 0/203 | 75/202 (37%) |
| SYP-3 | 12 | 0.97 | 0.54 | 0.54 | 2/213 | 109/213 (51%) |
| SYP-4 | 11 | 1.00 | 0.26 | 0.48 | 2/550 | 277/550 (50%) |
| SYP-5 | 10 | 1.00 | 0.99 | 0.59 | 2/554 | 173/554 (31%) |
| SMC-1 | 11 | 1.00 | 0.85 | 0.54 | 1/1300 | 805/1300 (62%) |
| SMC-3 | 12 | 1.00 | 0.51 | 0.41 | 0/1243 | 760/1243 (61%) |

SMC-1, SMC-3, and LEV-11, which harbor extensive coiled-coils, are highly conserved in *Caenorhabditis* (*Figure 1C and D*, see also *Surkont and Pereira-Leal, 2015*).

The role of the SC in regulating genetic exchanges, and consequently chromosome segregation, makes it a candidate for involvement in meiotic drive, where a genetic locus skews its own inheritance. Meiotic drive often incurs a fitness cost, creating pressure for the emergence of suppressors. This tit-for-tat evolutionary arms race leads to rapid evolution, which can be detected bioinformatically as positive selection. Indeed, meiotic drive has been invoked to explain the rapid evolution of SC proteins in *Drosophila (Hemmer and Blumenstiel, 2016)*. However, we found no evidence for positive selection in any *Caenorhabditis* SC protein (*Table 1*). Using the CodeML program from PAML (*Yang, 1997*), we found no significant difference between models M8a (no positive selection allowed, dN/dS ≤ 1) and model M8 (positive selection allowed). Consistent with the high divergence observed above (*Figure 1C and D*), we found that fewer than 50% of sites evolve under purifying selection when examined on a per-site basis using a Fixed Effects Likelihood model (*Table 1*). Our per-site analysis found almost no evidence of positive selection (no sites in SYP-1 or SYP-2 and only two sites each in SYP-3, -4, and -5; *Table 1* and *Figure 1—figure supplement 3A*). Altogether, our analysis indicates that neutral evolution (lack of constraint) explains the high divergence of SC proteins in the *Caenorhabditis* lineage.

## Protein length and coiled-coil domains are conserved in SC proteins

Despite the poor conservation of primary amino acid sequence in SC proteins (*Figure 1*, *Table 1*, and *Hemmer and Blumenstiel, 2016*), bioinformatic and functional analysis has pointed to the prevalence of coiled-coils (*Page and Hawley, 2004*). We used Paircoil2 to predict the likelihood that each position is part of a coiled-coil plotted throughout as a 'coiled-coil score' [1 – Paircoil2 score]; (*McDonnell et al., 2006*; *Figure 2A*). These plots reveal striking conservation of the position and length of coiled-coils.

For example, in SYP-1, the coiled-coil begins precisely at position 40 and ends at position 400 in all species (*Figure 2A*). Disruptions, observable as dips in an otherwise continuous coiled-coil, are also conserved (marked by asterisks in *Figure 2A*). SYP-1 has a short (<10 amino acids) disruption at position 110, whereas SYP-3 and SYP-5 have longer disruptions (20–50 amino acids). These disruptions might create bends in the otherwise rod-like structure of SC proteins (*Dunce et al., 2018*), similar to the conserved 'kinks' and 'elbows' in the coiled-coils of the kinetochore protein NDC80 (*Maure et al.,*

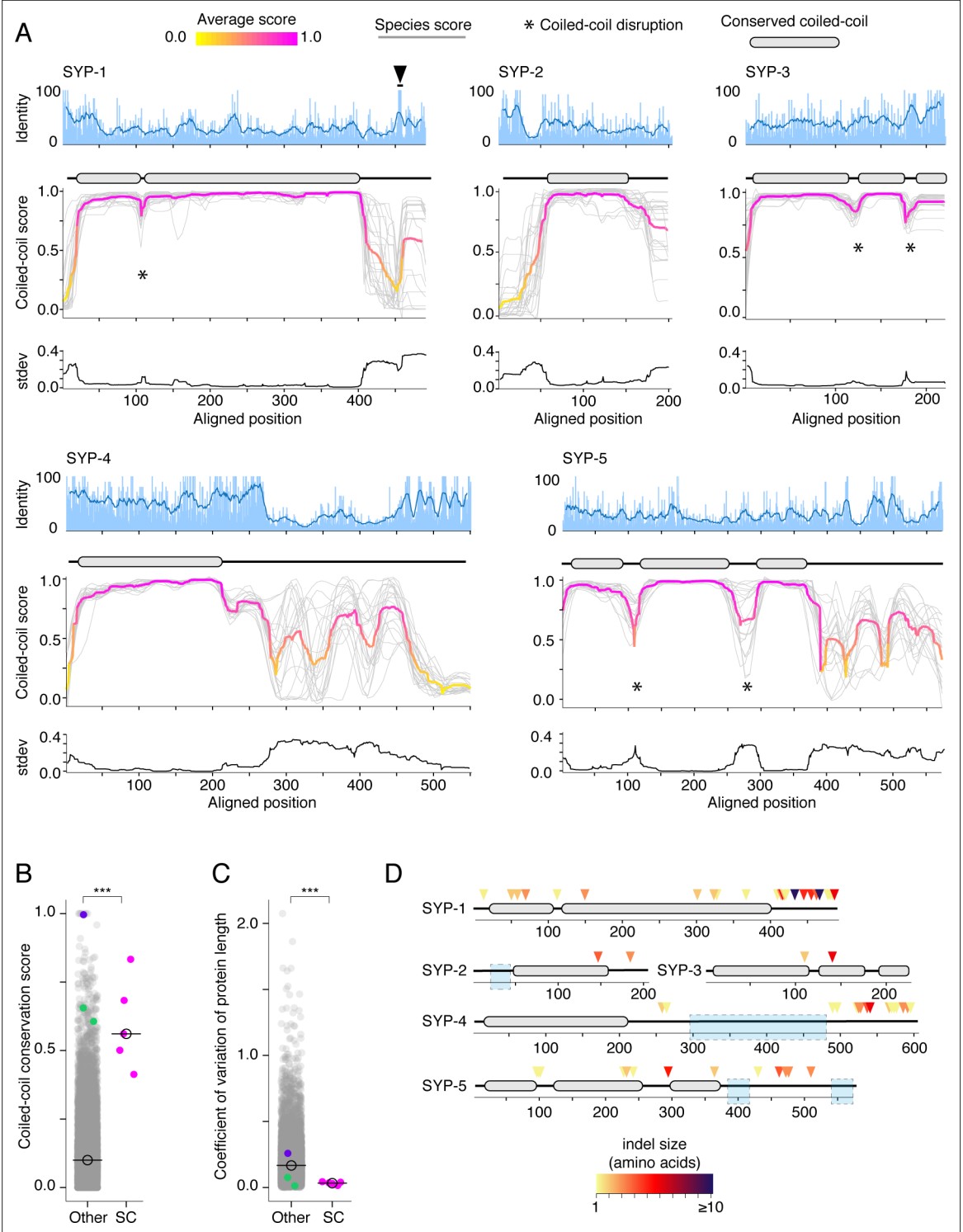

**Figure 2.** Protein length and coiled-coil domains are conserved features of synaptonemal complex proteins. (**A**) Percent amino acid identity (top), coiled-coil score (middle), and standard deviation of coiled-coil score (bottom) at each aligned position for SYP-1–SYP-5. Sliding window average of percent identity is shown (blue line). Coiled-coil conservation plots display the coiled-coil score (1 – Paircoil2 score) at each aligned position for all SYP proteins from each species (gray lines). Magenta and yellow lines, average score at each position. Gene models depicting conserved coiled-coil domains (gray-filled ovals) are shown above each coiled-coil conservation plot. The black arrowhead indicates the conserved Polo-Box Domain in SYP-1 (*Sato-Carlton et al., 2018*). Note that coiled-coil conservation is generally not reflected in elevated (or diminished) amino acid identity with the exception of SYP-4. (**B**) Dot plot comparing the coiled-coil conservation score of SC proteins to all other proteins in *Caenorhabditis*. The coiled-coil conservation score is the average minimum value of the coiled-coil score (1 – Paircoil2 score) for each position. Median coiled-coil conservation score

*Figure 2 continued on next page*

*Figure 2 continued*

for SC proteins = 0.56, median for all other proteins = 0.12. Wilcoxon rank sum test p-value=0.00019. (**C**) Dot plot comparing coefficient of variation of protein length of SC proteins to all other proteins in *Caenorhabditis*. Median for SC proteins = 0.04, median for all other proteins = 0.16. Wilcoxon rank sum test p-value=0.0004. SMC-1/3 (green) and LEV-11 (purple) are shown as controls in (**B**) and (**C**). LEV-11 (tropomyosin) coiled-coil conservation score of 0.996 is consistent with the importance of coiled-coils for its function (*Hitchcock-DeGregori, 2008*). (**D**) Indels in *Elegans* supergroup SC proteins. Gene model of SYP-1–SYP-5 coiled-coil domains with indel positions marked with colored arrowheads, with darker reds indicating larger indels. Light blue boxes surround regions that were excluded from analysis due to alignment uncertainty.

The online version of this article includes the following figure supplement(s) for figure 2:

**Source data 1.** Multiple sequence alignments used in indel analysis.

**Source data 2.** Phylogenetic trees used in indel analysis.

**Figure supplement 1.** Synaptonemal complex proteins have a significantly higher coiled-coil conservation score than other coiled-coil proteins.

**Figure supplement 2.** Synaptonemal complex proteins have limited regions of conserved disorder.

*2011*; *Hsu and Toda, 2011*) and the ring-like SMC-family proteins (*Yatskevich et al., 2019*; *Beasley et al., 2002*).

To quantitate the extent of coiled-coil conservation, we developed a scalar metric, the coiled-coil conservation score, that takes the minimum score (least likely to be part of a coiled-coil) from every aligned position (as in *Figure 2A*). This score is averaged across the alignment. Proteins with coiled-coils in the same position will have a higher score than proteins whose coiled-coils do not overlap or that lack extended coiled-coils altogether. Consistent with our qualitative observations, SC proteins have a significantly higher coiled-coil conservation scores on average compared to all other proteins (*Figure 2B*; median coiled-coil conservation score for SC proteins = 0.56, other = 0.12, p-value=0.00019) and compared to other coiled-coil proteins (*Figure 2—figure supplement 1*). This stands in contrast to their higher-than-average amino acid divergence (*Figure 1D*). Neither coiled-coils nor their edges leave discernible signatures of sequence conservation on SC proteins (*Figure 2A*). We also do not find strong correlation between the coiled-coils and amino acids under purifying selection (*Figure 1—figure supplement 3A*). These observations likely reflect the relatively loose requirements for coiled-coil formation: heptad repeats where the first and fourth amino acids are hydrophobic and the fifth and seventh amino acids are charged or polar (*Figure 1—figure supplement 3B*; *Truebestein and Leonard, 2016*). More broadly, this result suggests that the coiled-coil conservation score is more informative than a binary measure of coiled-coil domain prediction (i.e., presence/absence of coiled-coil). The additional information contained in the coiled-coil conservation score likely comes from the fact that it takes into account whether or not the coiled-coil domains are aligned across species, which reflects a higher degree of secondary structure conservation.

We wondered whether other secondary structural features are conserved in SC proteins. SC proteins often encode disordered domains of unknown function (*Zhang et al., 2020*). We used PONDR VL3 to predict the likelihood of disorder for all sites in *Caenorhabditis* SC proteins (*Figure 2—figure*

**Table 2.** Contingency table showing the number of alignment positions containing indels and lacking indels inside versus outside the coiled-coil domain of each synaptonemal complex protein. Two tailed p-value from Fisher's exact test is shown in the last column. Total number of insertions and deletions is depleted in the coiled-coil domains of SYP-1, SYP-4, and SYP-5.

| Protein | Inside coiled-coil | | Outside coiled-coil | | p-value |
|---|---|---|---|---|---|
| | Alignment positions with indels | Total alignment positions | Alignment positions with indels | Total alignment positions | |
| SYP-1 | 30 | 407 | 54 | 157 | <0.0001 |
| SYP-2 | 4 | 104 | 3 | 92 | N.A. |
| SYP-3 | 8 | 251 | 0 | 33 | N.A. |
| SYP-4 | 3 | 237 | 31 | 161 | <0.0001 |
| SYP-5 | 14 | 372 | 14 | 131 | 0.0063 |
| Total | 59 | 1371 | 102 | 574 | <0.0001 |

*supplement 2*). Unlike coiled-coils, the length and position of disordered domains were mostly varied between species. However, a few disordered regions were conserved, including the C-termini of SYP-1, -3, -4, and -5 and the N-terminus of SYP-2 (*Figure 2—figure supplement 2*). This analysis indicates that while multiple secondary structures might be under selection in SC proteins, conservation is particularly strong for coiled-coils.

Finally, we explored whether the conservation of coiled-coils in SC proteins is reflected in their overall length. We analyzed coefficient of variation of protein length of all *Caenorhabditis* proteins. We found that the median variation in length of SC proteins is significantly lower than that of other *Caenorhabditis* proteins (*Figure 2C*, median coefficient of variation of protein length for SC proteins = 0.03, other = 0.16, p-value=0.0004), again, in striking contrast to their diverged primary amino acid sequence (*Figure 1C and D*). Low variation of protein length suggests strong purifying selection against insertions and deletions (indels). Indeed, we find a significant depletion of indels in the coiled-coils of SC proteins compared with regions outside the coiled-coils (*Figure 2D*, *Table 2*; two-tailed

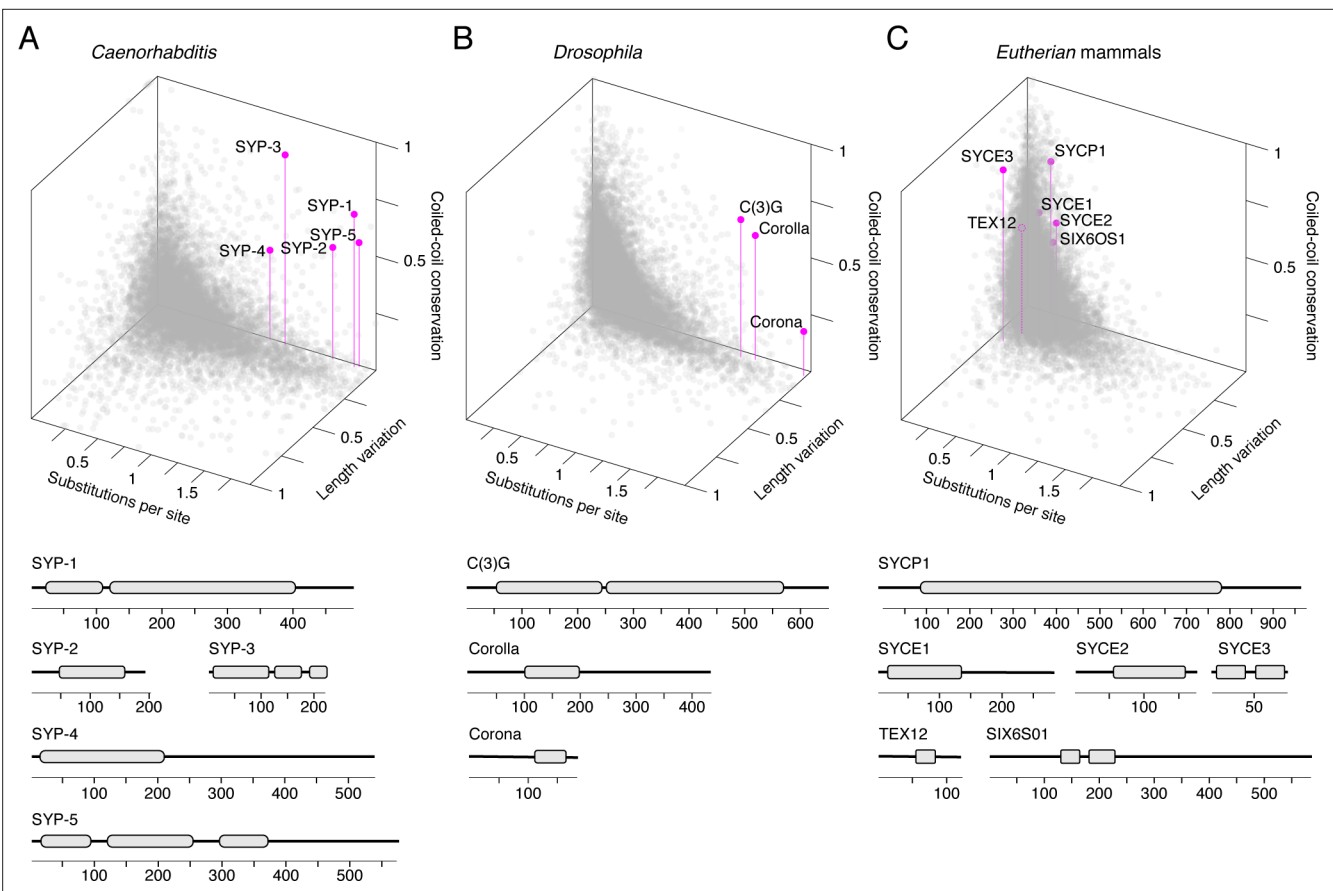

**Figure 3.** Synaptonemal complex proteins have an unconventional, but conserved, evolutionary signature. 3D scatter plot comparing amino acid substitutions per site, coefficient of variation of protein length, and coiled-coil conservation score of all proteins in 25 *Caenorhabditis* species (**A**), 30 *Drosophila* species (**B**), and 15 mammalian species (**C**). SC proteins, pink dots with vertical lines; other proteins, gray dots. Gene models of SC proteins depicting conserved coiled-coils derived from conservation plots (*Figure 2*, *Figure 3—figure supplements 2 and 3*) are shown below each scatter plot.

The online version of this article includes the following figure supplement(s) for figure 3:

**Source data 1.** Proteomes used to generate 3D scatter plots.

**Source data 2.** Genomic sequence of *L. africana* SYCE3 from UCSC Genome Browser.

**Figure supplement 1.** Synaptonemal complex proteins have an unconventional evolutionary signature.

**Figure supplement 2.** Coiled-coil plots for *Drosophila* synaptonemal complex proteins.

**Figure supplement 3.** Coiled-coil plots for mammalian synaptonemal complex proteins.

**Figure supplement 4.** Mammalian synaptonemal complex proteins display a similar, albeit weaker, evolutionary signature.

p-value from Fisher's test comparing alignment positions with indels to alignment positions that are part of coiled-coils < 0.0001). When examined individually, we detected significant depletion of indels in the coiled-coils of SYP-1, -4, and -5 (**Table 2**; SYP-2 and SYP-3 had only two indels each, preventing statistical analysis). In summary, selection acts against indels in the coiled-coils of the SC, consistent with the conservation of SC protein length and domain arrangement.

Taken together, these analyses highlight several conserved features that are not apparent from primary amino acid conservation alone. We find that SC proteins show an unusual evolutionary signature consisting of three key features: (1) high amino acid divergence, (2) conserved coiled-coils, and (3) low coefficient of variation of protein length. To demonstrate this point, we plotted these metrics for all proteins in *Caenorhabditis* on a Cartesian coordinate system (**Figure 3A**, **Figure 3—figure supplement 1A**). While most proteins clustered near the origin, SC proteins are among the few proteins situated away from it (**Figure 3A**, **Figure 3—figure supplement 1A**). Of the few proteins clustering with SC proteins are several that play a role in the mitotic spindle (**Figure 3—figure supplement 1B and C**; p-value=0.0076). Noteworthy among them is SPD-5, a component of the pericentrosomal material that nucleates spindle microtubules (**Figure 3—figure supplement 1C and D**; **Hamill et al., 2002**). Like SC proteins, SPD-5 and its apparent functional homologs in other eukaryotic clades – Cdk5Rap2 in vertebrates and Centrosomin in *Drosophila* – do not share significant sequence homology despite their conserved and essential functions in cell division (**Woodruff, 2018**; **Eisman and Kaufman, 2013**).

## The evolutionary signature of SC proteins is conserved across phyla

Next, we wondered whether this evolutionary signature was restricted to SC proteins in *Caenorhabditis*. In a dataset of 30 *Drosophila* species spanning 40 million years of evolution, the three known *Drosophila* SC proteins – Corolla, Corona, and the transverse filament protein C(3)G – exhibit a similar evolutionary signature to SC proteins in *Caenorhabditis*, and likewise, occupy a region of the coordinate system occupied by few other proteins (**Figure 3B**, **Figure 3—figure supplement 2**).

Analysis of proteomes of 15 mammalian species representing Xenarthra, Afrotheria, Laurasiatheria, and Euarchontoglires (*Eutherian* mammals, ~100 million years) revealed a similar, albeit weaker, evolutionary signature of the six SC proteins SYCP1, SYCE1–3, TEX12, and SIX6OS1 (**Figure 3C**, **Figure 3—figure supplements 3 and 4**). While mammalian SC proteins exhibited conserved coiled-coil domains (**Figure 3—figure supplements 3 and 4**), they had a lower overall divergence compared to *Caenorhabditis* and *Drosophila* SC proteins (median amino acid substitutions per site for mammalian SC proteins = 0.26 compared to 1.64 in *Caenorhabditis* and 1.69 in *Drosophila*). This might be explained by the overall lower median divergence of the proteome along the mammalian lineage (median amino acid substitution per site for all mammalian proteins = 0.068 compared to 0.43 in *Caenorhabditis* and 0.27 in *Drosophila*). Despite being relatively conserved, mammalian SC proteins do have a higher median amino acid substitutions per site than other proteins in mammals, although this comparison is not significant (**Figure 3—figure supplement 4A**). The relatively constrained divergence of mammalian SC proteins, including along their coiled-coils, might indicate novel functions adopted by these domains in mammals (**Fraune et al., 2012**; **Dunce et al., 2018**).

Protein length conservation was also not as apparent in mammalian SC proteins (**Figure 3—figure supplement 4C**). However, coefficient of variation of protein length is the metric most impacted by genome annotation errors (see Materials and methods). For example, the high coefficient of variation of SYCE3 is driven by conspicuous N- and C-terminal extensions in one species – *Loxodonta africana* (African elephant) (**Figure 3C**, **Figure 3—figure supplement 4C**). The conservation of a now internal start codon and the many sequence ambiguities around SYCE3 in the *L. africana* genome suggest that these extensions could be gene annotation errors (**Figure 3—source data 2**, see also Discussion). Despite these differences, mammalian SC proteins follow the same trend as SC proteins in *Drosophila* and *Caenorhabditis*, suggesting that these conserved features reflect structural and/or functional constraints acting on the SC.

## Identification of a novel SC protein in *Pristionchus pacificus*

Most SC proteins have been identified independently in each lineage by genetic and cell biological methods (**MacQueen et al., 2002**; **Hurlock et al., 2020**; **Colaiácovo et al., 2003**; **Smolikov et al., 2007**; **Smolikov et al., 2009**; **Page and Hawley, 2001**; **Sym et al., 1993**; **Dobson et al., 1994**). The strong evolutionary signature of SC proteins raised the possibility that we could identify SC proteins

*in silico* by relying on intra-clade conservation patterns. To test this, we turned to the nematode genus *Pristionchus,* which is distantly related to *Caenorhabditis*. While *P. pacificus* is an emerging model organism for evolutionary and developmental biology, its SC is poorly characterized. One SC protein, Ppa-SYP-4, has been identified based on a limited sequence similarity to the C-terminus of *C. elegans* SYP-4 (***Rillo-Bohn et al., 2021***). Notably, Ppa-SYP-4's coiled-coil is predicted to be only 31 nm long, too short to span the 100 nm SC as a head-to-head dimer, suggesting that we have yet to identify a transverse filament protein in *P. pacificus*. We developed a bioinformatics pipeline to identify SC proteins based on our prior analysis of *Caenorhabditis, Drosophila,* and mammals. Rather than leveraging sequence homology across distant genera, we categorized the proteome of eight sequenced *Pristionchus* species based on our evolutionary signature – high amino acid substitutions per site, low coefficient of variation of protein length, and high coiled-coil conservation scores – and generated a candidate list of *Pristionchus* SC proteins (***Figure 4A and B***). We further filtered our list for germline enriched genes (see Materials and methods) and were left with only eight candidate SC proteins, one of which was, gratifyingly, Ppa-SYP-4.

We generated null alleles in our top three candidates using CRISPR/Cas9 and found that one of these genes, PPA16075, is an SC protein that we named Ppa-SYP-1. Neither of the other two candidates (PPA10754 and PPA35551) exhibited meiotic phenotypes. Examination of condensed meiotic chromosomes (DAPI bodies) revealed that almost all chromosomes in *Ppa-syp-1* lacked crossovers (***Figure 4C***). Consistent with this, *Ppa-syp-1* mutant hermaphrodites give rise to almost no viable self-progeny; a reflection of embryonic aneuploidy caused by uncoordinated meiotic chromosome segregation (***Figure 4D***). To visualize Ppa-SYP-1, we raised an antibody against 18 amino acids in its C-terminus and generated two strains with internal tags: an HA tag near the N-terminus and an OLLAS tag near the C-terminus, with only minor effects on SC function (***Figure 4C and D***, ***Figure 4— figure supplement 1***, and Materials and methods). As expected for an SC protein, HA::Ppa-SYP-1 localized to the interface between the parental chromosomes in meiotic prophase (***Figure 4E***). Its C-terminus formed parallel tracks spaced 114 nm apart on average, whereas the N-terminus formed a single, narrower track (***Figure 4F–H***). This indicates that Ppa-SYP-1 is a transverse filament protein arranged in a head-to-head orientation, spanning the width of the SC. Ppa-SYP-1 appears to be a functional homolog of *C. elegans* SYP-1: they are both transverse filament proteins that contain similar coiled-coil domain structures, predicted to be 55 and 51 nm long, respectively (***Figure 4I***, compare to ***Figure 2A***, SYP-1; ***Tung and Roeder, 1998***; ***Steinert et al., 1993***), and both harbor Polo-Box Domains in their C-termini (STP at positions 451–453 for *C. elegans* SYP-1 [***Sato-Carlton et al., 2018***] and positions 658–660 and 681–683 for Ppa-SYP-1). This is despite the fact that Ppa-SYP-1 has no significant homology to *C. elegans* SYP-1, or, for that matter, to any known SC protein.

## Discussion

The discrepancy between the ultrastructural and functional conservation of the SC, on one hand, and the sequence divergence of its constituent proteins, on the other, have long baffled chromosome biologists. Here, we show that while analysis of per-site identity and similarity reveals high degree of sequence divergence, SC proteins exhibit significant and widespread conservation of protein length and of coiled-coil domains. SC proteins thus provide a clear example of a first-order amino acid fitness landscape tolerant of mutations accompanied by second-order fitness landscapes (properties derived from primary amino acid sequence) that impose strong evolutionary constraints. This provides an alternative framework to understand rapid divergence of proteins performing essential functions, in addition to the more well-documented processes of adaptive evolution and positive selection (***McLaughlin et al., 2017***).

The evolutionary constraints acting on SC proteins appear to be distinct from constraints imposed on other meiotic proteins. For example, the proteins that make up the meiotic axis, which acts as the assembly scaffold for the SC and includes the cohesin complex and meiotic HORMA domain proteins (HORMAD1 and HORMAD2 in mammals, HTP-1/2, HTP-3 and HIM-3 in *C. elegans*), are recognizable by sequence homology across all eukaryotic clades (***Cobbe and Heck, 2004***; ***Aravind and Koonin, 1998***; ***van Hooff et al., 2017***). This homology can largely be attributed to structured domains; the ATP binding domain in SMC proteins and the eponymous HORMA domain are defined by conserved amino acids that underlie their globular structure and proper fold. In the case of SC proteins, we hypothesize that primary sequence is less important as long as the basic requirements of a coiled-coil

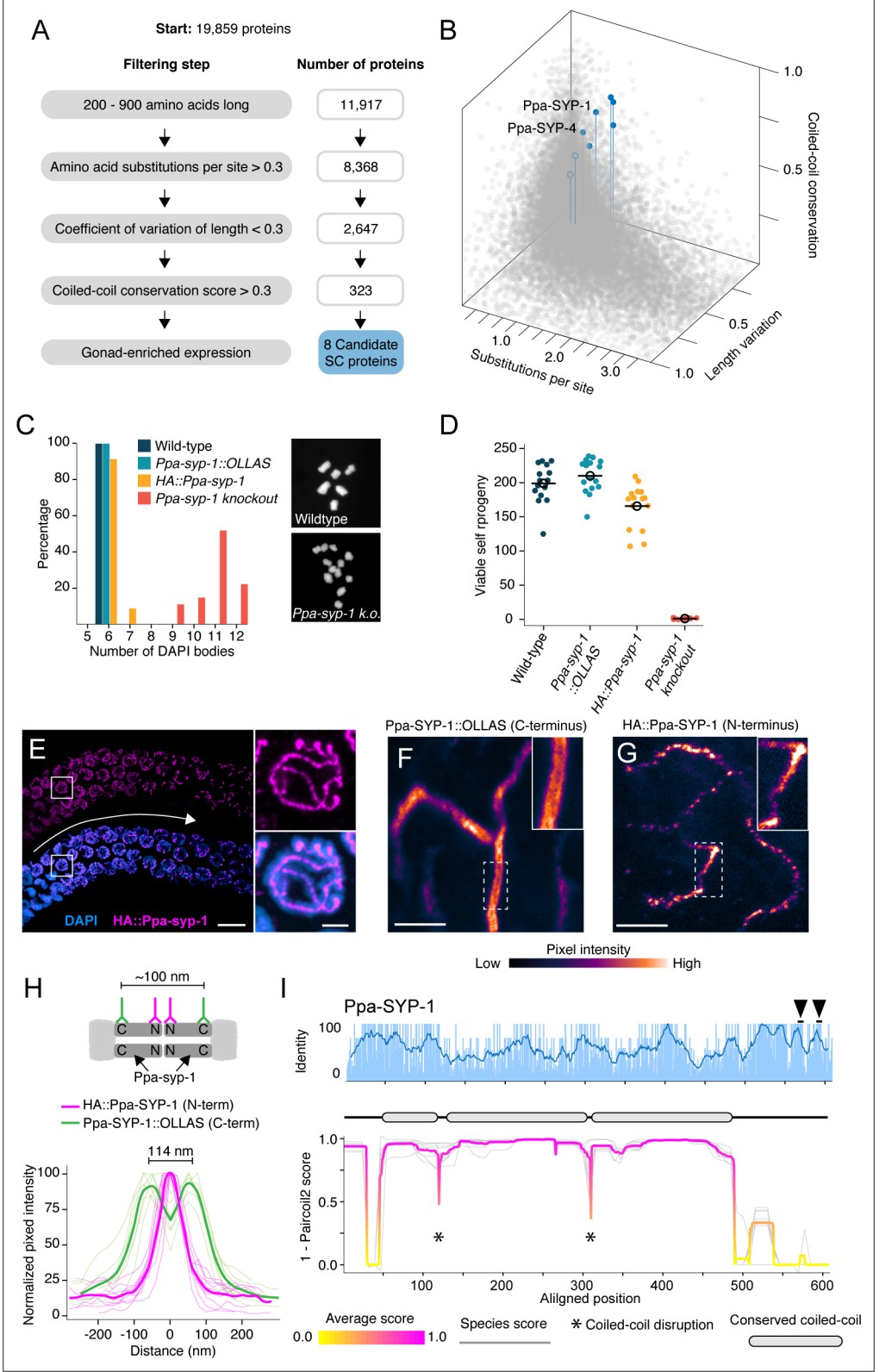

**Figure 4.** Identification of *P. pacificus* SYP-1. (**A**) Flow chart of filtering steps to identify candidate synaptonemal complex proteins in *Pristionchus*, with the number of remaining proteins after each step shown to the right. See Materials and methods for details. (**B**) 3D scatter plot comparing amino acid substitutions per site, coefficient of variation of protein length, and coiled-coil conservation score of all *Pristionchus* proteins. Blue dots, candidate SC

*Figure 4 continued on next page*

*Figure 4 continued*

proteins; other, gray dots. (**C, D**) Number of DAPI bodies (**C**) and total viable self-progeny (**D**) from wild-type *P. pacificus, Ppa-syp-1::OLLAS, HA::Ppa-syp-1,* and *Ppa-syp-1* knockout hermaphrodites. In (**C**), representative DAPI body images are shown for wild-type and *Ppa-syp-1* knockout. (**E**) Left: image of a prophase region of a gonad from a *HA::Ppa-syp-1* hermaphrodite where meiosis progresses from left to right (white arrow), stained with antibodies against HA (magenta). Scale bar = 5 µm, scale bar in inset = 2 µm. (**F, G**) STED images of a representative chromosome from *Ppa-syp-1::OLLAS* (**F**, C-terminus) and *HA::Ppa-syp-1* (**G**, N-terminus) hermaphrodites stained with antibodies against OLLAS or HA tags, respectively (colored according to pixel intensity). Insets show higher magnification of sections in the dashed boxes. Note the 'railroad tracks' configuration in (**F**). Scale bars in (**F, G**) = 1 µm. (**H**) Top: model depicting Ppa-SYP-1 as a transverse filament protein, with antibodies targeting the C- and N-termini. Expected distance between C-terminal epitopes is ~100 nm. N-terminal epitopes are expected to be too close to be resolved. Bottom: line scans of normalized pixel intensity across the SC in Ppa-SYP-1::OLLAS (green) and HA::Ppa-SYP-1 (magenta). Bold green and magenta lines represent the average of multiple line scans. (**I**) Average pairwise percent amino acid identity (top) and coiled-coil score (bottom) for Ppa-SYP-1 from eight *Pristionchus* species. Sliding window average of percent identity is shown (blue line). Observable steps in the percent identity bar graphs are attributed to fewer species in the *Pristionchus* proteome dataset (8 *Pristionchus* species vs. 25 *Caenorhabditis* species in *Figure 2*). Magenta and yellow lines in coiled-coil plot, average score at each position; gray lines, individual species scores. Gene model depicting conserved coiled-coil domains (gray-filled ovals) is shown above the coiled-coil plot. Black arrowheads point to two Polo-Box Domains in Ppa-SYP-1, which are conserved in *Pristionchus*.

The online version of this article includes the following figure supplement(s) for figure 4:

**Source data 1.** Sequences of CRISPR reagents.

**Source data 2.** Protein sequences of SYP-1 and SYP-2 candidates from *C. plicata* and *C. monodelphis*.

**Figure supplement 1.** Ppa-SYP-1 antibody.

**Figure supplement 2.** Candidate SYP-1 proteins in *C. plicata*.

**Figure supplement 3.** Candidate SYP-1 proteins in *C. monodelphis*.

**Figure supplement 4.** Candidate SYP-2 proteins in *C. monodelphis*.

---

domain of a certain length are met. Although the axis proteins have conserved globular domains, they have been shown to evolve faster than other proteins on average (*Dapper and Payseur, 2019*), potentially due to positive selection acting to modulate recombination rate (*Sandor et al., 2012*; *Johnston et al., 2018*).

In the SC, the strong selection acting on coiled-coils likely reflects biophysical and structural properties that are conserved across phyla. Coiled-coils in transverse filament proteins help determine the 100 nm space between the parental chromosomes (*Sym and Roeder, 1995*; *Billmyre et al., 2019*; *Ollinger et al., 2005*). However, this role by itself cannot explain the prevalence and conservation of coiled-coils in SC proteins that are not transverse filaments. Moreover, it cannot explain conserved disruptions in the coiled-coils (*Figure 2*). An attractive hypothesis is that the specific arrangement of the coiled-coils and their disruptions determines other aspects of the 3D architecture of the SC lattice (*Dunce et al., 2018*), such as its axial depth or the uniform lateral spacing between the ladder rungs (*Figure 1A*). In that case, many functions of the SC could be maintained by swapping endogenous coiled-coils with orthologous or synthetic ones of similar length and arrangement.

An alternative, non-mutually exclusive role for the coiled-coils is promoting phase-separation. Despite its ordered appearance, the SC in worms, flies, and yeast has recently been shown to assemble through phase-separation (also referred to as condensation; *Rog et al., 2017*). Constituent subunits of condensates, including SC proteins, can enter and exit condensates and move within them. Coiled-coils can facilitate phase-separation (*Schmidt and Görlich, 2015*; *Li et al., 2012*), potentially by promoting multivalent interactions (*Newton et al., 2021*; *Mitrea and Kriwacki, 2016*). This is consistent with the poor per-site conservation of SC proteins since multivalent interactions can rely on molecular features exhibited by groups of amino acids (e.g., charge or hydrophobicity) rather than tight, 'lock-and-key' interfaces formed by specific tertiary structures. Tellingly, SPD-5, which shares an evolutionary signature with SC proteins in *Caenorhabditis* (*Figure 3—figure supplement 1*), has been shown to promote microtubule nucleation in mitosis through condensation (*Woodruff et al., 2017*). Given the growing number of characterized condensates in the cell (*Banani et al., 2017*), it is tempting to speculate that many of their constituent proteins are subject to unconventional

evolutionary pressures that are not apparent in their primary amino acid sequence. Prime candidates to examine in this regard are proteins with conserved length and/or arrangement of disordered domains (*Figure 2—figure supplement 2*). Disordered protein domains can drive phase-separation (*Darling et al., 2018*; *Uversky, 2017*) and tend to evolve more rapidly than their ordered counterparts (*Brown et al., 2002*).

The wide applicability of our approach hinges on the availability of high-quality genomes deeply sampled within clades. New technologies to assemble genomes de novo and the ever-lower costs of sequencing are moving us quickly in this direction (*Sohn and Nam, 2018*). But even now, many eukaryotic clades are well represented including *Anopheles* mosquitoes (*Neafsey et al., 2015*), *Aspergillus* fungi (*Kjærbølling et al., 2020*), *Apicomplexans* (*Martínez-Ocampo, 2018*), and choanoflagelletes (*Richter et al., 2018*). In addition, insight gained from our work might allow *in silico* prediction of functional homologs in species lacking closely related sequenced species (*Figure 4—figure supplement 2*; *Figure 4—figure supplements 3 and 4*).

Despite rapidly evolving amino acid sequence, purifying selection acts to limit length variation in SC proteins (*Figures 2 and 3*). The indels that are present in SC proteins are depleted within the coiled-coils (*Figure 2D*, *Table 2*). Such uncoupling is unusual since indels and substitutions typically occur together (*Tóth-Petróczy and Tawfik, 2013*; *Leushkin et al., 2012*). In fact, selection acting on indels has been demonstrated only in a handful of cases. One such case are sperm ion channels in primates, rodents, and flies, where positive selection for indels yielded N-terminal tails of highly varied lengths (*Podlaha et al., 2005*; *Podlaha and Zhang, 2003*; *Cooper and Phadnis, 2017*). Robust genome-wide identification of indels is complicated by assembly and annotation errors that can be mistaken for indels. When manually annotating our dataset of SC proteins in *Caenorhabditis*, we corrected annotation errors in 18% of sequences, most of which would have been otherwise mistakenly scored as alterations to protein lengths (see Materials and methods). Future work to develop methods to test for selection against indels is likely to shed light on the evolutionary dynamics impacting protein length variation and on the mechanisms underlying them.

Our ability to detect the unconventional conservation of SC proteins relied on the ultrastructural conservation of the SC across eukaryotes. This knowledge was gained through the widespread application of electron microscopy and the serendipitous ability to observe the SC without any molecular knowledge of its constituent subunits. Unlike the SC, however, much of our current understanding of cellular organization relies on the application of molecular tools (e.g., antibodies, tagged transgenes). These efforts are often informed and directed by conservation of primary amino acid sequence to select 'interesting' targets for cell biological and genetic experiments, and to actively avoid so-called orphan genes. Our work shows that by focusing our explorations under the streetlamp that are BLAST searches we might be ignoring conserved cellular structures and consequential biological processes.

## Materials and methods

**Key resources table**

| Reagent type (species) or resource | Designation | Source or reference | Identifiers | Additional information |
|---|---|---|---|---|
| Gene (*Pristionchus pacificus*) | PPA16075; Ppa-syp-1 | El Paco genome reference, V2 | | |
| Gene (*P. pacificus*) | PPA10754 | El Paco genome reference, V2 | | |
| Gene (*P. pacificus*) | PPA35551 | El Paco genome reference, V2 | | |
| Gene (*Caenorhabditis elegans*) | syp-1 | https://wormbase.org/# 012-34-5, WS279 | F26D2.2 | See *Figure 1—source data 5* |
| Gene (*C. elegans*) | syp-2 | https://wormbase.org/# 012-34-5, WS279 | C24G6.1 | See *Figure 1—source data 5* |
| Gene (*C. elegans*) | syp-3 | https://wormbase.org/# 012-34-5, WS279 | F39H2.4 | See *Figure 1—source data 5* |

*Continued on next page*

*Continued*

| Reagent type (species) or resource | Designation | Source or reference | Identifiers | Additional information |
|---|---|---|---|---|
| Gene (*C. elegans*) | *syp-4* | https://wormbase.org/# 012-34-5, WS279 | H27M09.3 | See *Figure 1—source data 5* |
| Gene (*C. elegans*) | *syp-5* | https://wormbase.org/# 012-34-5, WS279 | Y54E10A.12 | See *Figure 1—source data 5* |
| Gene (*C. elegans*) | *spd-5* | https://wormbase.org/# 012-34-5, WS279 | F56A3.4 | |
| Strain, strain background (*P. pacificus*) | PS312 | Caenorhabditis Genetics Center | PS312 | |
| Genetic reagent (*P. pacificus*) | *Ppa-syp-1* | This paper | | Null allele, available by request from the Rog lab |
| Genetic reagent (*P. pacificus*) | *HA::Ppa-syp-1* | This paper | | In-frame insertion of HA tag, available by request from the Rog lab |
| Genetic reagent (*P. pacificus*) | *Ppa-syp-1::OLLAS* | This paper | | In-frame insertion of OLLAS tag, available by request from the Rog lab |
| Antibody | Anti-Ppa-SYP-1 (rabbit polyclonal) | This paper, Pocono Rabbit Farm and Laboratory | | Antibody targeting GSKSNKRQTRARGKKRTK in Ppa-SYP-1 Available by request from the Rog lab (1:1000) |
| Antibody | Anti-HA (mouse monoclonal) | Roche | 12CA5 | (1:500) |
| Antibody | Anti-OLLAS (rat monoclonal) | Invitrogen | MA5-16125 | (1:200) |
| Sequence-based reagent | Guide RNAs, DNA repair templates, and genotyping primers | This paper | | See *Figure 4—source data 1* |
| Sequence-based reagent | Alt-R CRISPR-Cas9 tracrRNA | Integrated DNA Technologies | Cat # 1072532 | |
| Peptide, recombinant protein | Antigen for anti-Ppa-SYP-1 antibody | This paper | | Peptide sequence: GSKSNKRQTRARGKK |
| Peptide, recombinant protein | Alt-R S.p. Cas9 Nuclease V3 | Integrated DNA Technologies | Cat # 1081058 | |

## Identification of SC proteins in *Caenorhabditis*

To identify SC proteins in *Caenorhabditis,* we used *C. elegans* SYP-1–SYP-5 as queries in BLASTP and tBLASTn searches of 19 species in the *Elegans* group and *Drosophilae* supergroup. For the remaining six species, which are more distantly related to *C. elegans,* we used SYP-1–SYP-5 sequences from all *Elegans* and *Drosophilae* supergroups species as BLAST queries. We compared the syntenic location (5′ and 3′ neighbor genes, *Figure 1—source data 1*) of each BLAST hit and built gene-specific phylogenies to confirm orthology (*Figure 1—figure supplement 2*, *Figure 1— source data 2*). In several cases (23/125, 18%), gene annotations were either absent or incorrect. For example, three annotations merged two genes and seven had obvious errors in intron/exon boundaries. Uncorrected, these annotations would have manifested as apparent indels. We corrected these errors manually using expression data when available and by alignment to closely related species (*Figure 1—source data 3*). In a few cases, ambiguities in genome assemblies prevented us from generating confident gene models. *C. japonica, C. inopinata, C. virilis, C. angaria,* and *C. monodelphis* SYP-4 all share significant homology to other SYP-4s. However, their gene models reside on short scaffolds, the edges of scaffolds, or contain ambiguous bases. Similarly, *C. tropicalis, C. waitukubuli, C. japonica,* and *C. angaria* SYP-5 all contained ambiguities. We scored each of these genes as present, but did not use them in further analyses. Alignments used for gene-specific phylogenies were generated using ClustalW (*Larkin et al., 2007*) implemented in Geneious

Prime (version 2019.0.4). Maximum likelihood phylogenies were generated with PhyML (version 3.3.20200621) with the LG amino acid substitution model and 100 replicates for bootstrap support (*Guindon et al., 2010*).

## Testing for positive selection

We selected the *Elegans* supergroup as an appropriate subset of species to test for recurrent positive selection (*McBee et al., 2015*). We used ClustalW as described above to make alignments of each SC protein and of SMC-1/3 (controls) from 12 *Elegans* group species (*C. zanzibari, C. tribulationis, C. sinica, C. briggsae, C. nigoni, C. remanei, C. latens, C. doughertyi, C. brenneri, C. tropicalis, C. inopinata,* and *C. elegans*). Since *C. elegans* contains a SYP-5 paralog, SYP-6, we excluded *C. elegans* SYP-5/6 from the SYP-5 protein alignment. We generated corresponding nucleotide alignments using Pal2Nal (*Suyama et al., 2006*). Each alignment, along with an *Elegans* group species tree (*Stevens et al., 2019*), was used as input to the CodeML sites model of PAML (*Yang, 1997 Figure 1—source data 4*). We compared models M1 (neutral) and M2 (selection), M7 (dN/dS < 1) and M8 (dN/dS < 1, plus an additional category of dN/dS > 1), and M8a (dN/dS ≤ 1) and M8. We tested for significance in each comparison using a likelihood ratio test. We ensured that our results were robust to codon substitution model and starting dN/dS by running each test with two codon models (F3 × 4 and the codon table derived from each alignment) and with two starting dN/dS values (dN/dS = 0.4 and dN/dS = 1.5). To ensure that our lack of detection of positive selection was not due to the relatively high divergence of the *Elegans* group species, we repeated the analysis excluding *C. elegans* and *C. inopinata,* the most divergent members of the *Elegans* species group. We found no evidence for positive selection using this less-diverged species set. We also tested for pervasive positive selection using a Fixed Effects Likelihood method (*Smith et al., 2015*) implemented at datamonkey.org. We used the same nucleotide alignments that we used for PAML analysis of the full *Elegans* group as input. Alignment-wide average pairwise percent identity for SYP-1–SYP-5 and SMC-1/3 (*Figure 1C*) and percent identity by site (*Figure 2A*) was calculated in Geneious. Sliding window percent identity (window size = 10 amino acids) was calculated in R (version 4.0.2).

## Generating orthogroups, making alignments, and calculating divergence

We used OrthoFinder (*Emms and Kelly, 2015*; *Emms and Kelly, 2019*) with default parameters to create groups of orthologous proteins (orthogroups) from *Drosophila, Eutherian* mammalian, and *Pristioncus* genomes or proteomes (*Figure 3—source data 1*). *Caenorhabditis* orthogroups were generated previously (*Stevens et al., 2019*). We removed paralogous proteins from each orthogroup by removing the species containing the duplicate gene from that orthogroup. For *Caenorhabditis, Drosophila,* and mammalian analyses, we only analyzed orthogroups that contained proteins from at least half of the possible species after removing paralogs (13, 15, and 7 species, respectively). In an effort to aid identification of SC proteins in *Pristionchus,* and since there are only eight *Pristionchus* genomes available, we did not apply these filtering steps to *Pristionchus* orthogroups. This resulted in 9924 *Caenorhabditis* orthogroups, 11,622 *Drosophila* orthogroups, 18,470 mammalian orthogroups, and 28,042 *Pristionchus* orthogroups. We aligned all orthogroups using ClustalW implemented in MEGA (*Kumar et al., 2018*; *Kumar et al., 2012*). We also used MEGA to estimate overall mean amino acid substitutions per site, calculated for all pairwise combinations, under a Poisson substitution model for each aligned orthogroup. We assumed rate variation among sites followed a gamma distribution with gamma parameter = 2.00.

## Coiled-coil domain prediction and coiled-coil conservation scores

We used Paircoil2 with window size = 28 for all coiled-coil domain predictions (*McDonnell et al., 2006*). To calculate the coiled-coil conservation score, we aligned the coiled-coil scores (1 – Paircoil2 score) for each orthogroup based on the amino acid alignment using a custom Python script. Alignment columns with fewer than 80% of species represented were removed in the plots shown throughout. For *Pristionchus* analyses, we removed alignment columns that had fewer than seven out of eight species represented. For proteins of interest (SYP-1–SYP-5, C(3)G, Corolla, Corona, SYCP1, SYCE1, SYCE2, SYCE3, TEX12, SIX60S1, and all candidate *Pristionchus* SC proteins), aligned coiled-coil scores were visualized using R. Average and standard deviation of aligned coiled-coil scores were

calculated and visualized in R. We then took the minimum value at each alignment position and averaged it across the length of the alignment. We refer to this averaged value as the coiled-coil conservation score. Conserved coiled-coils (gray ovals in *Figures 2A and 3*, *Figure 3—figure supplements 2 and 3*) were defined as regions of the coiled-coil plots at least 21 amino acids long where the average coiled-coil score was >0.8 and the standard deviation of the coiled-coil score was <0.1. To define a group of coiled-coil proteins for comparison to SC proteins (*Figure 2—figure supplement 1*), we selected orthogroups in which 90% of the proteins in the group had a coiled-coil domain of 21 amino acids or longer (916 *Caenorhabditis* orthogroups including all SC proteins).

## Coefficient of variation of protein length

Coefficient of variation of protein length was calculated as the standard deviation of protein length in each orthogroup divided by the mean length of the proteins in that group.

## Statistical tests comparing SC proteins to the proteome

We used a Wilcoxon rank sum test to compare median amino acid substitutions per site (*Figure 1D*), coiled-coil conservation score (*Figure 2B*), and coefficient of variation of protein length (*Figure 2C*) of SC proteins to the rest of the proteome.

## Disordered domain prediction

We used PONDR VL3 for all disordered domain predictions (*Peng et al., 2005*). We aligned PONDR VL3 scores for SYP-1–SYP-5 using a custom Python script and visualized the aligned scores using R. Average and standard deviation of the aligned PONDR VL3 scores were calculated and visualized in R.

## Indel analysis

We used ClustalW to generate an alignment of each SC protein (SYP-1–SYP-5) in the *Elegans* supergroup (*Figure 2—source data 1*). We constructed a neighbor-joining tree corresponding to each alignment (*Figure 2—source data 2*). We then scanned each alignment and manually counted instances of indels along the phylogeny. Each position in the alignment was then defined as either indel-containing or indel-lacking. Separately, each alignment position was classified as either part of or not part of a coiled-coil domain based on the coiled-coil score at each position in *C. elegans*. For the SYP-5 alignment, we excluded *C. elegans* SYP-5 because *C. elegans* contains a SYP-5 paralog, SYP-6. Therefore, we defined the coiled-coil alignment columns based on *C. briggsae* SYP-5. For statistical analysis, we generated a 2 × 2 contingency table comparing the prevalence of indel-containing positions to the coiled-coil domains and calculated two-sided p-values from Fisher's test from combined data from SYP-1 to SYP-5. Our null hypothesis was that indels would be equally likely to occur in coiled-coil and non-coiled-coil domains.

## 3D plots and hierarchical clustering

Proteins were assigned a point in a 3D Cartesian coordinate system where x = amino acid substitutions per site, y = coiled-coil conservation score, and z = coefficient of variation of protein length. For ease of viewing, we excluded the top 0.1% outliers from amino acid substitution per site and coefficient of variation of protein length from the *Caenorhabditis*, *Drosophila*, and *Eutherian* mammals 3D plots in *Figure 3*. Similarly, we excluded the top 1% of outliers from the same two categories in the *Pristionchus* 3D plot in *Figure 4*. We used all data points for *Pristionchus* SC candidate filtering (see below) and for *Caenorhabditis* proteome hierarchical clustering analysis. For hierarchical clustering analysis, we used the dist function in the stats package in R to generate a dissimilarity matrix based on Euclidean distance between each point. The dissimilarity matrix was used to perform complete linkage clustering using hclust, also in R. We plotted the results as a dendrogram with 15 leaves (*Figure 3—figure supplement 1*).

## Enrichment analysis

We used the Database for Annotation, Visualization and Integrated Discovery (DAVID v6.8) to test for enrichment of Gene Ontology (Biological Processes sub-ontology) in the two dendrogram leaves containing SC proteins (*Figure 3—figure supplement 1A*, blue and green dots). Significantly enriched categories were embryo development ending in birth or egg hatching (GO:0009792, p-value=$9.5 \times 10^{-9}$),

SC assembly (GO:0007130, p-value=0.0035), meiotic nuclear division (GO:0007126, p-value=0.0046), meiotic chromosome segregation (GO:0045132, p-value=0.0074), and mitotic spindle organization (GO:0007052, p-value=0.0079). p-Values reported were Benjamini corrected.

## *P. pacificus* maintenance

*P. pacificus* strains were grown at 20°C on NGM agar with *Escherichia coli* OP50. *P. pacificus* strain PS312 (obtained from the CGC) was used for injections and as a wild-type control. All strains were homozygous except for the *Ppa-syp-1* knockout, which was maintained as heterozygote. In order to perform progeny counts and DAPI body counts of homozygous *Ppa-syp-1* knockout animals (experiments described in more detail below), we singled animals to identify heterozygotes and identified the homozygous knockout animals among their progeny based on the presence of laid eggs but no viable progeny (compared to their wild-type or heterozygous siblings that had many viable progeny). These homogyzous knockout animals were genotyped by PCR and used for subsequent experiments.

## Prioritizing SC candidates for knockout in *P. pacificus*

We prioritized our list of *Pristionchus* candidate SC proteins using the following criteria: (1) the gene must be single copy in *P. pacificus*. (2) The protein must not have a significant BLAST match to any protein in *C. elegans*. Since *C. elegans* SC proteins have no significant homology to any *P. pacificus* protein, we reasoned that the reverse would also be true. (3) We prioritized candidates that showed gonad-specific enrichment based on RNA tomography (*Rödelsperger et al., 2021*). However, since the RNA tomography study *Rödelsperger et al., 2021* used gene names from previous strand-specific transcriptome assemblies (*Rödelsperger et al., 2016*; *Rödelsperger et al., 2018*) and our analysis was based on the El Paco V2 genome assembly, we first identified candidates whose expression was enriched in J4 versus J1 *P. pacificus* using RNAseq data (NCBI SRA: PRJNA628502, Michael Werner, personal communication). This left 68 adult enriched candidates. We used each of these 68 candidates as queries in a BLAST search against the strand-specific transcriptome to identify a gene name that corresponds to the RNA-tomography data and selected genes that were gonad enriched. We were left with eight candidates, including Ppa-SYP-4 (*Rillo-Bohn et al., 2021*).

## Construction of *P. pacificus* strains

*P. pacificus* J4 animals were grown for 24 hr at 20°C prior to injection. Injection mix was prepared as follows: to make the RNA mix, we combined 4 µl tracrRNA (IDT, 200 µM) and 4 µl crRNA (200 µM) and incubated at 95°C for 5 min. We let the RNA mix cool on the benchtop for 5 min. To make the injection mix, we combined 3.5 µl RNA mix, 1 µl Cas9 protein (IDT, 10 µg/µl), 3 µl single-stranded DNA repair template (200 µM), and 1 µM DNA duplex buffer (IDT). To detect CRISPR events, we screened the pooled F1 progeny of injected $P_0$ animals for the loss of an endogenous restriction site near the Cas9 cut site (so-called 'jackpot' plates). We then singled F1s from jackpot plates for subsequent genetic analysis.

Construction of null mutants in candidates was done by inserting a premature stop codon in the N-terminus of each candidate. Stop codons were generated either via homology directed repair or by random indels generated by Cas9. *Ppa-syp-1* had a stop codon 94 bp from the N-terminal 'ATG,' *PPA10754* had a 10 bp insertion 6 bp from the N-terminus, resulting in a frame shift and a stop codon after 75 bp, and *PPA35551* had a N-terminal deletion of 222 bp, resulting in a stop codon after 78 bp. Construction of Ppa-SYP-1 strains tagged with OLLAS and HA was done by inserting each tag within the *Ppa-syp-1* coding sequence using homology directed repair. The HA tag was inserted 5′ of the extended coiled-coil and is flanked by three glycine linkers on each side. The OLLAS tag was inserted 3′ of the extended coiled-coil. Both tagged strains are homozygous viable and produce similar numbers of viable self-progeny to wild-type *P. pacificus* (*Figure 4D*). All edits were verified by Sanger sequencing. See *Figure 4—source data 1* for a list of gene-specific crRNAs, primers, and restriction enzymes used for genotyping.

## Progeny counts

We picked at least 10 single *P. pacificus* J4 animals onto individual plates (wild-type n = 15, *HA::Ppa-SYP-1* n = 15, *Ppa-SYP-1::OLLAS* n = 16, *Ppa-syp-1* knockout n = 11). We transferred each animal to

a new plate every day for 5 days. The resulting progeny on each plate were counted as adults 4 days after the parent was moved off of the plate. Progeny counts were performed at 20°C.

## Ppa-SYP-1 antibody development

We raised an antibody against the 18 C-terminal-most residues in Ppa-SYP-1 (residues 688–705, GSKSNKRQTRARGKKRTK). Pocono Rabbit Farm and Laboratory immunized two rabbits (#37938 and #37939) four times over a 70-day period (70-day antibody production package). Data presented here were generated using unpurified serum from the exsanguination of rabbit #37938. We confirmed the specificity of the antibody by staining *Ppa-syp-1* mutant animals and noticed only background nuclear signal (*Figure 4—figure supplement 1*).

## Imaging

We dissected age-matched *P. pacificus* hermaphrodites (24 hr post-J4) in 30 µl 1× Egg Buffer, essentially as described in *Phillips et al., 2009* with 0.01% Tween-20 and 0.005% tetramisole on a 22 × 22 mm coverslip. To fix, we added equal volume of a 2% formaldehyde solution in 1× Egg Buffer and incubated for 1 min. We removed nearly all dissection/fixation solutions from the sample and picked up the coverslip with a HistoBond microscope slide (VWR). Samples were then frozen on dry ice. After freezing, we snapped the coverslip off and immediately immersed samples in –20°C methanol for 1 min. We then washed samples 3 × 5 min in PBST (0.1% Tween-20) and incubated in primary antibodies overnight at 4°C. We washed samples 3 × 5 min in PBST and incubated in secondary antibodies for 2 hr at room temperature. Samples were washed for 10 min in PBST and for 10 min in DAPI (5 µg/µl). Samples were mounted in NPG-glycerol. Antibodies were used as follows: (primaries) mouse anti-HA (Roche 12CA5) 1:500, rat anti-OLLAS clone L2 (Invitrogen) 1:200, rabbit anti-Ppa-SYP-1 (Pocono Rabbit Farm) 1:1000, (secondaries for confocal) donkey anti-mouse Alexa 488 (Jackson ImmunoResearch) 1:500, donkey anti-rabbit Alexa 488 (Jackson ImmunoResearch) 1:500, (secondaries for STED) goat anti-mouse STAR RED (Aberrior) 1:100, and donkey anti-rat Alexa 594 (Jackson ImmunoResearch 1:500). For STED microscopy, samples were prepared as above except we omitted DAPI staining and mounted in Aberrior Mount liquid antifade (Aberrior) instead of NPG-glycerol. STED images were acquired with an Aberrior STEDYCON mounted on a Nikon Eclipse Ti microscope with a 100 × 1.45 NA oil objective. Confocal images were acquired on a Zeiss LSM880 with Airyscan and a 63 × 1.4 NA oil objective. STED images are a single z-section, and confocal images are partial maximum intensity projections.

## DAPI body counting

48 hr post-J4, *P. pacificus* hermaphrodites were dissected and stained as described above, except for omitting antibody staining. Oocytes with condensed chromosomes (typically in the –1 or –2 position) were imaged with a confocal z-stack. DAPI bodies were counted from 3D renderings in Zen Blue.

## Line scan measurements

Pixel intensities from STED images of *Ppa-syp-1:OLLAS* (n = 13 chromosomes) and *HA::Ppa-syp-1* (n = 10 chromosomes) were measured via line scan perpendicularly across the SC in ImageJ (version 2.1.0/1.53c). Pixel intensities were normalized to the maximum value in each line scan. Line scans from multiple chromosomes were aligned using the center of the SC as a reference. Line scan averages were calculated in R.

## Identifying SYP candidates in *C. plicata* and *C. monodelphis*

To identify candidate SYP-1 proteins in *C. plicata* and *C. monodelphis,* we examined orthogroups that contained *C. plicata* and/or *C. monodelphis* proteins and lacked any species where we previously identified SYP-1. We used Paircoil2 to predict coiled-coil domains in all remaining proteins and noted the longest coiled-coil domain in each protein (the longest stretch of Paircoil2 scores below 0.1, allowing for one or two amino acids to exceed the threshold value without causing the domain to end). We chose proteins that had (1) coiled-coil domains of at least 150 amino acids long, (2) total protein length between 300 and 1300 amino acids, and (3) no significant BLAST matches in *C. elegans* as SYP-1 candidates. We performed a similar analysis to identify candidate SYP-2 proteins in *C. monodelphis*. We examined orthogroups that contained a *C. monodelphis* protein and lacked any species

where we previously identified SYP-2. We predicted coiled-coil domains in all remaining proteins using Paircoil2 and noted the longest coiled-coil domain in each protein. We chose proteins that had (1) coiled-coil domains between 21 and 150 amino acids long, (2) total protein length between 175 and 225 amino acids, and (3) no significant BLAST matches in *C. elegans* as SYP-2 candidates.

## Acknowledgements

We thank Michael Werner for *P. pacificus* advice, reagents, and RNAseq data, Lewis Stevens for *Caenorhabditis* orthogroups, The University of Utah Center for High Performance Computing for computational resources, The University of Utah Cell Imaging facility for STED microscopy resources, and the Rog lab, Harmit Malik lab, Sophie Caron, Jon Seger, and Fred Adler for discussions in various stages of this project. We would also like to thank Yuval Mazor, Nitin Phadnis, Talia Karasov, and Michael Werner for critical reading of the manuscript, and Sara Nakielny for comments on the manuscript and editorial work. Worm strains were provided by the CGC, which is funded by NIH Office of Research Infrastructure Programs (P40 OD010440).

## Additional information

### Funding

| Funder | Grant reference number | Author |
|---|---|---|
| Eunice Kennedy Shriver National Institute of Child Health and Human Development | T32HD007491 | Lisa E Kursel |
| National Institute of General Medical Sciences | R35GM128804 | Ofer Rog |

The funders had no role in study design, data collection and interpretation, or the decision to submit the work for publication.

### Author contributions

Lisa E Kursel, Ofer Rog, Conceptualization, Data curation, Formal analysis, Funding acquisition, Investigation, Methodology, Supervision, Validation, Visualization, Writing - original draft, Writing - review and editing; Henry D Cope, Investigation

### Author ORCIDs

Lisa E Kursel (iD) http://orcid.org/0000-0002-1178-8230
Ofer Rog (iD) http://orcid.org/0000-0001-6558-6194

### Decision letter and Author response

Decision letter https://doi.org/10.7554/eLife.72061.sa1
Author response https://doi.org/10.7554/eLife.72061.sa2

## Additional files

### Supplementary files

• Transparent reporting form

### Data availability

All data generated or analyzed during this study are included in the manuscript and supporting files. Source data files have been provided for all figures.

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
