## [Editor Report]

Although the synaptonemal complex is an essential, deeply conserved structure that holds meiotic chromosomes together, the constituent proteins evolve exceptionally rapidly. This rapid evolution has hindered the identification of synaptonemal complex proteins based solely on sequence homology. This article overcomes this challenge by developing and validating a clever protein structure-based approach that leverages sequence divergence – rather than sequence conservation – to identify novel synaptonemal complex components.

---

## [Decision Letter]

**Decision letter after peer review:**

Thank you for submitting your article "Unconventional conservation reveals structure-function relationships in the synaptonemal complex" for consideration by *eLife*. Your article has been reviewed by 3 peer reviewers, and the evaluation has been overseen by a Reviewing Editor and Jessica Tyler as the Senior Editor. The reviewers have opted to remain anonymous.

The reviewers appreciated the creative and powerful approach implemented to discover novel SC proteins. The experimental verification of one such protein in Pristionchus was especially compelling.

Essential revisions:

1) The intra-clade conservation pattern approach applied to Pristionchus should also be applied to *C. plicata* and *C. monodelphis*, both of which lack SYP-1 (and one, SYP-2) by PSI-BLAST. Note that functional validation of any candidates is not expected.

2) Additional justification for inferring that eutherian mammals exhibit the same striking patterns shown in figures 3A and 3B as *Drosophila* and Caenorhabditis should be provided. The SYP proteins shown in 3C plot appear quite different than those in plots in 3A and 3B. Providing (in the supplement) plots of each of the three parameters independently for eutherian mammals -- like plots in 1D, 2B and 2C – might offer more compelling evidence to justify that the pattern is "similar." If instead the pattern is actually restricted to Caenorhabditis and *Drosophila*, then the authors might speculate on what would be a biologically interesting distinction.

3) Further justification for the conservation metric is warranted. The observation that the three control genes containing a coiled-coil domain are not different than the SC proteins suggests that the conservation parameter is not any more informative than simply the domain itself. A supplemental figure to 2B that contrasts SC proteins with all proteins that have a coiled-coil domain would be one way to convince the reader of the power of the conservation metric.

4) Please include a discussion (and possibly an analysis, if relevant) of the lateral/axial element proteins to offer important context for the focus on central proteins. The reader is left with the impression that lateral proteins are all strictly conserved (like the examples offered – SMC1 and SMC3). If indeed all lateral/axial element proteins are conserved, please offer one or two hypotheses for why. If some of these proteins instead evolve rapidly, can the method be developed for identifying central proteins also identify lateral proteins?

5) Clarify the relevance of the INDEL parameter to specific SYP proteins, highlighting the point that INDELs are not universal to SYP proteins.

---

## [Author Response]

Essential revisions:1) The intra-clade conservation pattern approach applied to Pristionchus should also be applied to C. plicata and C. monodelphis, both of which lack SYP-1 (and one, SYP-2) by PSI-BLAST. Note that functional validation of any candidates is not expected.

The lack of enough species with sequenced genomes closely related to *C. plicata* and *C. monodelphis* makes a direct application of our method, which relies on intra-clade diversity, essentially impossible. Indeed, we were able to apply the method to Pristionchus because there are eight publicly-available sequenced genomes. Although a few sister species to *C. monodelphis* have been identified (*C. sonorae*, *C. sp 57*, and *C. auriculariae*), only the genome of *C. auriculariae* is currently available, making it impossible to deduce patterns of intra-clade divergence.

Despite the fact that we could not apply our method, as is, to individual genomes, we were able to identify SYP-1 and SYP-2 candidates by applying our new understanding of conserved secondary structure in SC proteins. We searched for proteins that had (1) No significant homology to proteins in *C. elegans*, and (2) Coiled-coil domains of the expected length for SYP-1 and SYP-2. We identified four SYP-1 candidates in *C. plicata*, and five SYP-1 and 21 SYP-2 candidates in *C. monodelphis*. While we didn’t use it as a requirement, we placed SYP-1 candidate with Polo-like kinase binding motifs (‘STP’) in their C-terminus at the top of our lists since this motif was conserved in the other Caenorhabditis SYP-1 and in Ppa-SYP-1. We have added details of our analysis to the Methods section (“Identifying SYP candidates in *C. plicata* and *C. monodelphis*”) and figures displaying the coiled-coil prediction plots for all candidates (Figure 4 —figure supplement 2, 3 and 4).

2) Additional justification for inferring that eutherian mammals exhibit the same striking patterns shown in figures 3A and 3B as *Drosophila* and Caenorhabditis should be provided. The SYP proteins shown in 3C plot appear quite different than those in plots in 3A and 3B. Providing (in the supplement) plots of each of the three parameters independently for eutherian mammals -- like plots in 1D, 2B and 2C – might offer more compelling evidence to justify that the pattern is "similar." If instead the pattern is actually restricted to Caenorhabditis and Drosophila, then the authors might speculate on what would be a biologically interesting distinction.

As suggested, we have added dot plots comparing mammalian SC proteins to all other mammalian proteins for the three metrics central to this manuscript – amino acid substitutions per site, coiled-coil conservation scores and coefficient of variation of protein length. The plots can be found in Figure 3 —figure supplement 4.

These plots provide additional evidence that the evolutionary pattern of mammalian SC proteins is similar to (although weaker than) that of Caenorhabitis and *Drosophila*.

In panel (A), we show the median amino acid substitutions per site of SC proteins is higher than other proteins in mammals, although the difference is not significant. We discuss two reasons why the divergence trend is weaker for mammalian SC proteins in the results. Briefly summarized they are, 1. The overall divergence of the mammalian proteome is less than that of the *Caenorhabditis* or *Drosophila* proteome, and 2. Mammalian SC proteins may face additional evolutionary constraints due to novel functions including mammalian-specific protein interactions.

In panel (B), we show that mammalian SC proteins have a significantly higher coiled-coil conservation score than other proteins.

In panel (C), we show coefficient of variation of protein length for mammalian SC proteins is not significantly different than other proteins. We hypothesize that this could be due to gene annotation errors which plague even very high-quality genomes. For example, we found annotation errors in 23 (18%) of the 125 *Caenorhabditis* SC proteins examined in this study. Uncorrected, these errors often read as large insertions or deletions, and artificially large coefficient of variation. We use *L. africana* SYCE3 to demonstrate how potential annotation errors could impact our measure of length variation in mammalian SC proteins. *L. africana* SYCE3 has conspicuous N- and C-terminal extensions not found in any other SYCE3. Excluding that single protein – *L. africana* SYCE3 – reduces the average length variation from 29% to 4% in the SYCE3 orthogroup, below the median of other proteins. Correspondingly, the median SC coefficient of variation of protein length drops from 20% (unfilled black circle) to 12% (dashed, unfilled circle). While systematic manual annotation of the Eutherian mammals proteomes is beyond the scope of this manuscript, we added in the Discussion explicit reference to the implications of annotation errors on our ability to systematically address evolutionary pressures affecting indels.

3) Further justification for the conservation metric is warranted. The observation that the three control genes containing a coiled-coil domain are not different than the SC proteins suggests that the conservation parameter is not any more informative than simply the domain itself. A supplemental figure to 2B that contrasts SC proteins with all proteins that have a coiled-coil domain would be one way to convince the reader of the power of the conservation metric.

We thank the reviewers for this important suggestion. Indeed, the inclusion of the few examples in Figure 2 were meant as demonstration rather than a statistical analysis. To create a group of proteins that would serve as appropriate control for conservation of the length and organization of the of coiled-coils, we selected orthogroups in which 90% of the proteins had a coiled-coil domain of 21 amino acids or longer. This left 916 *Caenorhabditis* orthogroups including all SC proteins. We found that the median coiled-coil conservation score of SC proteins was significantly higher than that of the other coiled-coil proteins, confirming our comparisons to the entire proteome. We have included this analysis as a figure supplement to Figure 2 (dot plot shown here and Figure 2 —figure supplement 1) and added text to the Results and Methods describing the analysis.

More broadly, this result suggests that our coiled-coil conservation score is more informative than a binary measure of coiled-coil domain prediction (i.e. presence/absence of coiled-coil). The additional information contained in the coiled-coil conservation score likely comes from the fact that we take into account whether or not the coiled-coil domains are aligned across species; which reflects a higher degree of secondary structure conservation. We believe that future work to develop better measures of conservation of secondary structures will hone our ability to identify conservation of other protein classes.

4) Please include a discussion (and possibly an analysis, if relevant) of the lateral/axial element proteins to offer important context for the focus on central proteins. The reader is left with the impression that lateral proteins are all strictly conserved (like the examples offered – SMC1 and SMC3). If indeed all lateral/axial element proteins are conserved, please offer one or two hypotheses for why. If some of these proteins instead evolve rapidly, can the method be developed for identifying central proteins also identify lateral proteins?

We have added two paragraphs to the Discussion in our revised manuscript where we address the evolution of the axis proteins and contextualize our focus on the central region proteins. In brief, unlike the central region proteins, lateral/axis proteins can be identified by sequence homology across distant taxa. Key components of the axis are conserved from fungi to mammals, including homologs of the budding yeast proteins Hop1p and Rec8p. This is because they contain conserved structured domains (HORMA domain, for example) that have specific primary sequence requirements. In contrast, SC proteins have less strict sequence requirements because their core structural feature, coiled-coil domains, can be made from multiple combinations of amino acids. Since most studies of protein evolution and conservation have focused on cases, like the lateral/axis proteins, where primary sequence and structural conservation go hand-in-hand, we argue that it is valuable to also examine proteins whose structural conservation is not reflected in primary sequence. We have also clarified that positive selection can play a role in either framework.

5) Clarify the relevance of the INDEL parameter to specific SYP proteins, highlighting the point that INDELs are not universal to SYP proteins.

We have clarified this point in our revised manuscript, highlighting that when analyzed as a group, indels are excluded in coiled-coils of *Caenorhabditis* SC proteins, and that significance is also observed for specific SC proteins where enough indels are present to perform statistical tests. Two of the SC proteins, SYP-2 and SYP-3, had only two indels each, preventing us from performing tests of significance. We have also added text to the Discussion directly addressing the limitations of automatically-assigned gene annotations on the ability to test evolutionary pressures on indels genome-wide.